# Harnessing Artificial Intelligence to Enhance Global Breast Cancer Care: A Scoping Review of Applications, Outcomes, and Challenges

**DOI:** 10.3390/cancers17020197

**Published:** 2025-01-09

**Authors:** Jolene Li Ling Chia, George Shiyao He, Kee Yuen Ngiam, Mikael Hartman, Qin Xiang Ng, Serene Si Ning Goh

**Affiliations:** 1NUS Yong Loo Lin School of Medicine, National University of Singapore, 10 Medical Dr. S117597, Singapore 119077, Singaporee0902066@u.nus.edu (G.S.H.); 2Department of Surgery, National University Hospital, Singapore 119074, Singapore; surnky@nus.edu.sg (K.Y.N.); ephbamh@nus.edu.sg (M.H.); 3Saw Swee Hock School of Public Health, National University of Singapore and National University Health System, 12 Science Drive 2, #10-01, Singapore 117549, Singapore; 4SingHealth Duke-NUS Global Health Institute, Singapore 169857, Singapore

**Keywords:** breast cancer, Artificial Intelligence, machine learning, early detection, diagnostic imaging, clinical decision support, scoping review

## Abstract

Breast cancer remains a leading health challenge worldwide, particularly in low-resource settings where access to early detection and treatment is limited. Artificial Intelligence (AI) offers promising solutions to enhance breast cancer care by improving early detection, streamlining diagnosis, optimizing treatment planning, and supporting healthcare providers in clinical decision-making. This study provides a comprehensive scoping review of the applications, benefits, and challenges of using AI in breast cancer care. The findings highlight how AI can bridge gaps in global healthcare systems, particularly in underserved regions, by increasing efficiency, reducing costs, and enhancing accuracy in diagnosis and treatment. By identifying key trends and addressing barriers to AI adoption, we hope to guide the development of practical, patient-focused AI solutions and encourage further exploration of its potential and to inspire innovation and collaboration within the research community to reduce disparities in breast cancer outcomes globally.

## 1. Introduction

Breast cancer remains a critical health issue worldwide, with consistently high incidence rates that place an enormous burden on healthcare systems. This burden is especially pronounced in low- and middle-income countries, where resources for early screening, diagnosis, and treatment are often limited. Incidence rates vary significantly, from under 40 cases per 100,000 women in some Asian and African regions to over 80 per 100,000 in high-income regions such as Australia, New Zealand, North America, and parts of Europe [1]. In addition to the high rates of occurrence, breast cancer leads to substantial disability-adjusted life years (DALYs) lost compared to other cancers [2], underlining an urgent need for innovative solutions to improve early detection, diagnosis, and treatment.

Despite advances in medical technology and breast cancer care, significant inequities persist, with high-income countries achieving far better outcomes than low-resource settings [3]. These disparities stem from differences in healthcare infrastructure, access to screening programs, and patient awareness and literacy levels [4]. The World Health Organization (WHO) launched the Global Breast Cancer Initiative (GBCI) in 2021 to address these gaps, setting ambitious targets such as ensuring early detection for at least 60% of invasive breast cancers, securing a diagnosis within 60 days of initial consultation, and achieving an 80% treatment completion rate with favorable outcomes [5,6]. However, achieving these goals in low-resource settings remains challenging due to factors such as restricted access to mammography and a shortage of specialists, which contribute to later-stage diagnoses in regions like Sub-Saharan Africa [7,8].

To surmount these challenges, AI offers promising solutions by interpreting breast imaging rapidly, integrating well with portable diagnostic devices, and enhancing imaging accuracy, which supports breast cancer screening and diagnosis in regions lacking mammography access and specialists like radiologists and pathologists [9,10,11]. Additionally, AI can improve remote diagnostics through telemedicine, enabling specialists to provide guidance without requiring patients to travel long distances [9]. AI also contributes to capacity building by providing training resources for local healthcare workers [12]. In areas lacking multidisciplinary tumor boards (MTBs), AI can address some of the gaps in comprehensive, standardized, and personalized treatment planning. Finally, AI’s ability to analyze large, complex datasets supports predictive analytics, optimizing resource allocation to improve timely and accurate breast cancer diagnosis and treatment in underserved communities, may potentially help to reduce global health disparities [12,13].

With this background in mind, this scoping review aims to (1) comprehensively examine the landscape of AI in breast cancer management, (2) map the evolution and integration of AI technologies in clinical settings, and (3) study the potential roles AI can play in enhancing diagnostic, therapeutic, and management strategies for global health. By focusing on diagnostic applications in imaging and pathology, innovations in treatment, and overarching management strategies, this review aims to shed light on both the potential and the challenges of leveraging AI to make a substantial impact on global breast cancer care.

## 2. Methods

This scoping review was designed to map the existing literature on AI applications in breast cancer management, identify key themes, and highlight gaps that warrant further research [14]. This scoping review protocol adhered to the PRISMA-ScR (Preferred Reporting Items for Systematic Reviews and Meta-Analyses extension for Scoping Reviews) guidelines [15] and the methodological framework outlined by Arksey and O’Malley [14], as well as further recommendations made by Levac et al. [16].

A comprehensive search strategy was developed to identify relevant studies; further information about the search terms can be found in Appendix A. Briefly, the PubMed, Web of Science, Cochrane Library, and Embase databases were searched from inception to May 2024. Keywords used included “Artificial Intelligence” and “Breast Cancer” and their equivalent Medical Subject Heading (MeSH) terms. No restrictions were initially applied regarding geographic location, date, or language. In addition to peer-reviewed articles, a grey literature search was conducted using Google to identify preprints and unpublished studies with substantial data, specifically non-peer-reviewed evidence, such as government reports, organizational white papers, or preprint studies, which may be significant in the rapidly evolving field of AI in healthcare. This approach aimed to capture emerging insights and ongoing research not yet published in traditional journals. Preprints with robust data and relevance to AI in breast cancer care were included in the final review to ensure a comprehensive synthesis of the field.

Two independent researchers (J.L.L.C. and G.S.H.) reviewed the titles and abstracts of the retrieved articles based on the predefined inclusion and exclusion criteria, with any discrepancies resolved through discussion with a senior author (Q.X.N. or S.S.N.G.). Full-text articles and preprints of potentially relevant studies were subsequently assessed for eligibility. Studies were included if they (1) focused on AI applications in breast cancer care and (2) presented original data. Exclusion criteria encompassed abstracts, editorials, commentaries, duplicate publications, and articles without an English translation.

Following the selection of eligible studies related to AI and breast cancer, a structured data extraction process was followed to ensure consistency and comprehensiveness across the reviewed literature. Two independent reviewers (J.L.L.C. and G.S.H.) extracted relevant data from each included study. The key variables extracted included study characteristics; population details; type of AI and specific algorithms or models used; application focus; outcome measures; implementation barriers and facilitators, etc. All extracted data were organized into a structured charting table, providing a visual overview of the study characteristics and findings. This charting table was iteratively refined to capture emerging themes and allow for consistent categorization of AI applications, outcomes, and implementation factors. Each study was assigned to one or more thematic categories based on its primary AI focus area, enabling an organized synthesis of evidence. Two independent researchers (J.L.L.C. and G.S.H.) manually reviewed the extracted data to identify recurring patterns and topics, and this process involved iterative discussion and consensus building with the senior authors (Q.X.N. and S.S.N.G.) to name and label the themes. The charting table facilitated a thematic synthesis of the results, allowing us to identify key trends, gaps, and variations in AI applications for breast cancer care.

The data synthesis for this scoping review followed three stages to comprehensively overview the field and identify research gaps and emerging trends, adhering to Popay et al.’s best practices for narrative synthesis [17]. The first stage involved creating an evidence map to visually represent the distribution and characteristics of included studies, highlighting thematic concentrations and underrepresented areas. In the second stage, we analyzed the evidence map and synthesized data to pinpoint gaps in the existing research landscape. The final stage involved a narrative synthesis to integrate findings from individual studies into a cohesive interpretation, starting with a preliminary synthesis and exploring relationships within and between studies to identify patterns and trends. This ensured a robust synthesis of diverse study types and outcomes [18], with a close focus on breast cancer care applications, outcomes, and challenges globally.

## 3. Results

From an initial screen of 4334 articles, a total of 145 full-texts were reviewed, before 84 articles [18,19,20,21,22,23,24,25,26,27,28,29,30,31,32,33,34,35,36,37,38,39,40,41,42,43,44,45,46,47,48,49,50,51,52,53,54,55,56,57,58,59,60,61,62,63,64,65,66,67,68,69,70,71,72,73,74,75,76,77,78,79,80,81,82,83,84,85,86,87,88,89,90,91,92,93,94,95,96,97,98,99,100,101] were finally included in the review (Figure 1 details the article selection process). The characteristics of the studies reviewed are briefly summarized in Table 1. The majority were conducted in developed countries (*n* = 54). The majority of publications were in the last 10 years (*n* = 83).

The findings from the included studies were grouped into six main themes based on their primary focus area: AI for breast cancer screening, AI for image detection of nodal status, AI-assisted histopathology, AI in assessing post-neoadjuvant chemotherapy (NACT) response, AI in breast cancer margin assessment, and AI as a clinical decision support tool. This thematic grouping was intended to highlight the breadth of AI applications across different stages of breast cancer care and the groups are reflected within Table 1.

### 3.1. Key Findings and Themes

Based on the findings (Table 1), six main themes for AI applications were generated, namely, AI for breast cancer screening (*n* = 32), AI for image-detection of nodal status (*n* = 7), AI-assisted histopathology (*n* = 8), AI in assessing post-NACT response (*n* = 23), AI in breast cancer margin assessment (*n* = 5), and AI as a clinical decision support tool (*n* = 9). For breast cancer screening, AI applications are utilized on mammography and digital tomosynthesis, which creates a 3-dimensional picture of the breast using radiographs. Ultrasound image segmentation techniques are employed to evaluate axillary lymph node metastases. AI-supported histopathology encompasses functions such as diagnosis, categorization, and grading. AI-based predictions of post-NACT responses were done using MRI images and AI-based vacuum-assisted biopsy. Margin assessment can be enhanced with AI-assisted optical coherence tomography. AI has been used in clinical decision support tools to augment treatment decisions for complex breast cancer and in multidisciplinary tumor board settings.

Overall, the use of AI in healthcare, particularly in the early detection and treatment of breast cancer, has the potential to significantly reduce healthcare costs by shifting the focus from expensive, late-stage treatments to more affordable, early-stage interventions [103,104]. Figure 2 illustrates a timeline of the evolution of AI applications in breast cancer care.

### 3.2. AI Diagnostic Applications

AI has become a transformative tool in breast cancer screening and diagnosis, revolutionizing how medical professionals interpret and analyze diagnostic data. Early AI systems, such as the first-generation computer-assisted detection and diagnosis (CAD) tools approved by the FDA in 1998, achieved limited success, as evidenced by the Digital Mammographic Imaging Screening Trial (DMIST) in 2014 [29]. However, advancements in technology and increased research in machine learning and deep learning since 2012 have led to the development of new deep learning-based CAD systems which aim to address the limitations of their predecessors and improve diagnostic accuracy. Recent research on these systems has focused primarily on mammography, generating binary outcomes (benign versus malignant) and analyzing extensive datasets from numerous screening exams to maximize clinical impact and improve mortality rates [42].

Through the integration of radiomics, machine learning, and deep learning, AI has the potential to detect and diagnose breast cancer at earlier stages. Radiomics, which extracts quantitative data from medical images, helps identify subtle patterns and anomalies in breast imaging [105]. Machine learning (ML) algorithms can then analyze these radiomic features, alongside large clinical datasets, to generate accurate predictions. Deep learning (DL), with its layered artificial neural networks, further enhances the diagnostic process by classifying and recognizing images [106].

Numerous retrospective studies have shown promising results for AI-supported mammography in breast cancer screening, particularly in detecting and classifying malignancy risk with accuracy comparable to or better than that of radiologists. AI’s strength lies in its ability to detect suspicious findings, identify true negatives, and classify screenings as high-risk, potentially identifying interval cancers before they become clinically evident. For example, Lang et al. demonstrated that AI could detect interval cancers from previous screening exams with minimal signs of malignancy, which may contribute to reducing the rates of interval cancers and late-stage diagnoses. The Mammography Screening with Artificial Intelligence Trial (MASAI), a recent randomized controlled trial, also found that AI-assisted mammography screening achieved comparable cancer detection rates to traditional double reading while significantly reducing the workload [54].

AI has also been applied to Digital Breast Tomosynthesis (DBT) [107], approved by the FDA in 2011, for lesion detection, segmentation, and characterization. Notable studies include those by Pöhlmann et al. [76] and Fan et al. [41], who developed three-dimensional segmentation methods for mass detection. Fan et al. demonstrated high sensitivity (90%) across various subgroups using the 3D-Mask RCNN CAD system [41]. More recently, Ye et al. [94] used a dilated deep convolutional neural network approach for the automated segmentation of mass regions in DBT images, achieving an accuracy and sensitivity of 86.3%. However, these findings are limited by small sample sizes and warrant further evaluation to validate their effectiveness.

There has been an increasing body of research focused on the role of AI in diagnosing breast cancer axillary lymph node metastasis. Lymph node metastasis from primary breast cancer is common and its status is crucial in determining management and overall prognosis. With trials examining the possibility of safely omitting sentinel lymph node biopsy in patients with small primary tumors [108], it highlights the pressing need for non-invasive techniques to detect lymph node metastasis in early-stage breast cancer. AI techniques such as ML and DL offer promising solutions to improve and supplement diagnostic procedures. Sun et al. showed success in using convolutional neural networks in predicting axillary lymph node metastasis from primary breast cancer ultrasound images with high accuracy (72.6%), sensitivity (65.5%), and specificity (78.9%) [87]. Comparing the application of a back-propagation neural network (BPNN) AI algorithm-based ultrasound image segmentation technology to routine ultrasound diagnosis, Zhang et al. [96] showed BPNN algorithm is of high accuracy, sensitivity, and specificity for ultrasound image segmentation with a better segmentation effect. Results showed that the area under the two-dimensional receiver operating characteristic curve of the BPNN AI algorithm model classification was always higher than the area under the curve of manual segmentation, with segmentation accuracy highest at two hidden layer nodes with corresponding segmentation accuracy of 97.3% [96]. While further studies are warranted to comprehensively assess the utilization of AI for assisting in the identification of axillary lymph node metastasis, the existing research indicates the promising potential for AI algorithms to play a pivotal role in improving nodal assessment in the near future.

Furthermore, AI has been employed to improve the accuracy and efficiency of axillary lymph node assessment, with focus on identifying micrometastasis, a task challenging to detect manually. Lymph Node Assistant, or LYNA, a DL-based algorithm developed and validated by Liu et al. [63], achieved higher tumor level sensitivity and comparable slide performance to pathologists. Furthermore, Steiner et al. [85] showed that DL-assisted pathologists demonstrated higher accuracy (91% vs. 83%, *p* = 0.02) in detection of micrometastases in lymph nodes than pathologists alone. Steiner et al. also showed that the average review per image was significantly shorter with assistance. CAMELYON16, a researcher challenge competition, illustrated that some DLMs achieved better diagnostic performance (AUC 0.885 vs. 0.808) in identifying presence of micrometastasis than expert pathologists [22]. By comprehensively assessing each tissue patch on a slide, DL algorithms enhance sensitivity and specificity, reducing the risk of missed micrometastasis [63]. This improved detection can help guide treatment decisions, minimizing unnecessary axillary clearance and its associated morbidity [109].

Related to this, the traditional approach of manual interpretation of pathology slides is not only time-consuming but also subject to interobserver variability [110]. The advent of digital pathology slides has transformed histopathological slides into high-magnification digital images, facilitating the application of computer-based image analysis with AI algorithms. The integration of AI assists pathologists in identifying subtle features of specific breast cancer subtypes, thereby enhancing the accuracy of diagnosis and prediction of prognosis, allowing clinicians to make well-informed treatment decisions. Recent research has focused on formulating ML and DL algorithms aimed at detection and classification of breast cancers [31,46]. Han et al. [46] explored the use of a class structure-based deep convolutional neural network (CSDCNN) model in the multi-class classification of breast cancer. It achieved reliable and accurate recognition rates, with the ability to distinguish between eight subtypes of benign (adenosis, fibroadenoma, phyllodes tumor, and tubular adenoma) and malignant tumors (ductal carcinoma, lobular carcinoma, mucinous carcinoma, and papillary carcinoma) with 93.2% accuracy [46]. As biomarker status determination is one of the key elements of breast cancer diagnosis, carrying implications for treatment options, several studies have also explored the use of AI in the quantitative measurement of immunohistochemical stained estrogen receptor (ER), progesterone receptor (PR) and human epidermal growth factor receptor (HER2) and Ki67 images. With the use of DLM, Couture et al. [31] achieved 84% accuracy in predicting ER status. Meanwhile, Li et al. [58] successfully identified Ki-67 status with high accuracy and results very consistent with the gold standard (ICC > 0.9) and good interobserver repeatability. Skaland et al. [111] found a 100% concordance between HER2/neu expression scores with digital image analysis and conventional and modified FISH scores. This highlights the potential for AI to serve as a tool for the detection and classification of breast carcinoma, alleviating the workload of pathologists and enhancing the efficiency of clinical diagnosis.

As the grade of breast carcinoma is strongly correlated to prognosis and outcome, numerous studies have investigated the use of AI to improve the accuracy of histological grading assessment, with a focus in the assessment of mitotic activity [40]. Applying deep learning in image analysis, Couture et al. [31] was able to distinguish between low-, intermediate-, and high-grade carcinoma with 82% accuracy. Nateghi et al. [69] demonstrated an enhancement in tumor proliferation prediction accuracy with a proposed fully automated system assessing mitotic count using deep learning to select regions of high mitotic activity and then detect mitosis from the selected areas. Additionally, AI algorithms have been designed to provide quantitative measurements of nuclear morphology, with the potential for application across various tumor subtypes. Whitney et al. [92] showed that computer-extracted nuclear morphology features were able to predict risk categories for early stage ER+ breast cancers at 75–86% accuracy.

Given that tumor-infiltrating lymphocytes (TILs) are associated with better response to therapy and overall survival [112], several studies [21,80] investigated the application of deep learning to assist pathologists in evaluating TILs. The computerized image-based detection and grading of lymphocytic infiltration in HER2+ breast cancer pathology proposed by Basavanhally et al. [21] successfully discerned between high and low levels of TIL with an accuracy exceeding 90%.

### 3.3. AI for Treatment Innovations

In recent decades, there has been a significant shift towards less invasive treatment options for patients with breast cancer. Whereas radical mastectomy and axillary lymph node dissection were once the standard practices, breast-conserving surgery followed by radiotherapy is now an option for patients with early-stage breast cancer and is also a viable option for women with locally advanced breast cancer who show a favorable response to neoadjuvant therapies [113]. Furthermore, the high rates of pathologic complete response (pCR) following neoadjuvant chemotherapy in HER2-positive breast cancer subtypes raise the question of whether surgery could be safely avoided, especially if the local tumor has been effectively eradicated by neoadjuvant chemotherapy [114]. This could potentially minimize the complications related to surgery and anesthesia, thereby enhancing cosmetic outcomes and improving patients’ quality of life. However, it remains uncertain how to accurately identify patients who have no residual cancer after neoadjuvant chemotherapy and who can safely forgo surgical intervention [115]. Therefore, clinical research in the last 10 years has been mainly focused on imaging and minimally invasive biopsies to exclude residual cancer. While imaging can offer an assessment of the general response to NACT, it may not be sufficient to conclusively rule out or confirm residual disease after NACT [116,117]. Several studies [118,119] that investigated the use of image-guided, minimally invasive biopsies to exclude residual cancer showed high false negative rates, with the MICRA trial presenting a false negative rate of 37% [119]. Recent studies using AI have demonstrated promising outcomes in identifying patients eligible for de-escalated locoregional treatment.

AI has also been used to assess post-neoadjuvant chemotherapy (NACT) response, primarily centered on Magnetic Resonance Imaging (MRI). Numerous studies [73,118] have employed radiomics to predict pCR after NACT. Pesapane et al. [73] evaluated the accuracy of extracted radiomics features from highly homogenous breast MRI images and demonstrated that MRI-based radiomic features improved pretreatment prediction of pCR to NACT in addition to biological characteristics (AUC 0.83 (0.73–0.92)). In recent years, deep learning convolutional neural networks (CNN) [52] have gained increased application in predicting pCR in breast cancer patients undergoing NACT. Deep learning CNNs can analyze whole-breast MRI images without requiring annotations or tumor segmentation by radiologists, suggesting the potential to reduce workload and enable the analysis of multiple findings. Dammu et al. [32] employed a novel deep learning CNN to predict pCR in breast cancer patients treated with NACT, showing high accuracy (0.80 ± 0.03) and an AUC of 0.83 ± 0.03. These studies highlight the capacity of AI to identify patients with pCR eligible for de-escalated locoregional treatment.

Pfob et al. [74] devised a multimodal machine learning algorithm-based model combining patient, tumor, imaging, and biopsy variables known as “intelligent vacuum-associated biopsy (VAB)”. The AI model was aimed at identifying breast cancer patients with a pCR response to NACT. The validation set revealed a false negative rate of 0% (95% CI: 0–13.7%) for detecting residual cancer, a specificity of 40% (95% CI: 19.1–63.9%), and an AUC of 0.91 (0.82–0.97). Moreover, the AI-guided VAB performed better than imaging NACT alone, VAB alone, or combinations of both, thereby suggesting its capacity to identify patients with no residual disease who are eligible for the omission of breast and axillary surgery [74].

In terms of breast cancer margin assessment, which is very important for surgical planning, existing techniques for intraoperative margin assessment such as frozen section analysis, gross assessment, and optical coherence tomography (OCT) are limited in their accuracy, reporting speed, or both, impeding effective clinical management [119]. Novel technologies tailored for intraoperative use during breast-conserving surgery (BCS) focus on integrating wide-field OCT (WF-OCT) systems with computer vision and machine learning. This integration aims to streamline the AI-facilitated analysis of extensive imaging data generated by WF-OCT for real-time margin visualization. Levy et al. [56] evaluated the efficacy of an AI-facilitated WF-OCT system and demonstrated that the DL model accurately identified 96.8% of pathology-positive margins with an AUROC of 0.976 and AUPRC of 0.812, suggesting the potential to improve reported re-excision rates due to positive margins from around 20% to below the 5% mark. Other studies also demonstrated that automated DL algorithms for detecting lymph node metastasis in Hematoxylin and Eosin (H&E) stains of breast cancer patients performed significantly better than pathologists when faced with time constraints for diagnosis [22].

### 3.4. AI Applications in Management Strategies

Research is also being conducted on integrating Large Language Models (LLMs), such as AI-driven chatbots, into breast cancer management. Studies comparing recommendations by existing Large Language Models with breast cancer tumor board recommendations have yielded promising results. In one study, seven out of ten cases of ChatGPT-3.5’s recommendations were found to be similar to the final decision of the tumor board [84]. Another found a 58.8% concordance rate for preoperative recommendations for invasive breast cancer and noted that although it has potential, it still is not advanced enough to provide specific recommendations for breast cancer treatment [44]. The literature search yielded six studies [28,44,47,78,84,102] which noted the highest accuracy ranging from 88 to 98% for information extraction and question-answering tasks in relation to breast cancer diagnosis and management, but accuracy variations and the presence of erroneous outputs mean that subsequent validation or human oversight is still required.

## 4. Discussion

This scoping review mapped out the existing range of clinical applications for AI in breast cancer management, including diagnostic imaging, histopathology, treatment planning, and clinical decision support. The incorporation of AI in breast cancer care represents a transformative opportunity to reduce global disparities, especially in LMICs. The ability of AI to enhance diagnostic precision, streamline workflows, and provide real-time support to clinicians makes it a powerful tool for improving care and breast cancer outcomes [120,121,122].

As late-stage breast cancer typically has a poorer prognosis compared to early-stage cases [123], the iBreastExam (iBE) is an example of an innovative device leveraging emerging AI technologies which offers a cost-effective alternative to conventional mammography for screening [124]. It has been positioned as a valuable adjunct for breast cancer detection in LMICs with constrained healthcare budgets; in a study conducted in India involving 989 healthy women undergoing annual health checks, the iBE demonstrated a 19% higher sensitivity than traditional clinical breast examination for detecting breast lesions, along with impressive specificity (94%) and negative predictive value (98%) [125]. However, despite such success stories of AI in breast cancer screening, realizing their full potential and seamlessly integrating the technology into LMIC settings demands the careful consideration of multiple factors.

First, access to the necessary infrastructure is a critical barrier in LMICs. Many regions lack adequate imaging facilities, digital pathology systems, and reliable internet connectivity, which are essential for deploying AI technologies [7,11,126]. Establishing this infrastructure will require significant investment and collaboration among governments, private sectors, and international organizations [5,127]. Additionally, ensuring the availability of high-quality, locally relevant data is crucial for training AI models that accurately reflect the unique needs and challenges of these populations [128].

Second, workforce readiness is a key enabler of successful AI implementation. Healthcare providers need training to understand, trust, and effectively use AI tools. Without such capacity-building efforts, even the most advanced AI systems may face resistance or underutilization [7,128]. Public–private partnerships can play a pivotal role in creating training programs and supporting the integration of AI into clinical workflows [128].

Third, ethical and regulatory considerations must be addressed to ensure the safe and equitable use of AI. These include establishing frameworks for data privacy, algorithm transparency, and accountability [129,130,131]. Inherent biases in training data stemming from the under-representation of minority populations can result in algorithms learning and perpetuating such biases, further contributing to disparities in patient care and reducing the accuracies of AI [130,132]. Hence, AI systems must be designed and implemented with fairness in mind to avoid perpetuating these existing healthcare inequalities. Collaborative efforts involving policymakers, clinicians, and AI developers are essential to develop standards that promote trust and equitable access [5,127].

Finally, it is essential to align AI applications with patient-centered outcomes. While many studies have focused on improving diagnostic accuracy or operational efficiency, future research should prioritize the direct impact on patient health and well-being [7,127]. Long-term studies are needed to evaluate whether AI-enhanced care translates into better survival rates, improved quality of life, and reduced healthcare costs [103,104].

By addressing these challenges, AI has the potential to bridge healthcare gaps, particularly in resource-constrained settings. This review highlights the importance of fostering innovation while ensuring that the benefits of AI reach all patients equitably. Achieving this balance requires sustained commitment from the global healthcare and research communities to harness AI responsibly and inclusively [11,129].

### Limitations of Review

The key strength of this systematic review lies in its comprehensive overview of AI applications in breast cancer care, emphasizing how these technologies can improve diagnostic imaging, histopathology, and clinical decision support. It underscores AI’s potential to enhance early detection and streamline workflows, which may be particularly advantageous in resource-limited settings. However, this review also acknowledges limitations, such as the variability in AI models, which often stems from differences in the datasets used for training and validation. Many models are developed using data from high-income countries, which may not reflect the genetic, epidemiological, and healthcare-specific characteristics of populations in LMICs. This raises concerns about the generalizability and reliability of these models when applied across diverse clinical settings. Additionally, the lack of standardized outcome measures complicates the comparison of AI systems across studies, making it difficult to establish benchmarks for performance and effectiveness. Without clear metrics, it becomes challenging to identify which models offer the most benefit in real-world applications. Another limitation is the scarcity of long-term impact data, which are important for understanding the sustainability and safety of AI implementation in breast cancer care. Most studies focus on short-term diagnostic accuracy or workflow efficiency, with limited evidence on how AI integration affects patient outcomes, healthcare costs, and system-level dynamics over time. This lack of longitudinal evidence creates uncertainty for healthcare providers and policymakers considering large-scale adoption. Moreover, the reviewed papers do not discuss how AI technologies might exacerbate existing disparities in healthcare. High development and deployment costs could limit access to these innovations, leaving low-income countries, rural regions, and marginalized communities behind.

## 5. Conclusions

In summary, AI has demonstrated a wide range of clinical applications in breast cancer care, including diagnostic imaging, histopathology, treatment planning, and clinical decision support. These advancements offer promising opportunities to address global disparities in healthcare access and improve patient outcomes. However, the widespread adoption of AI is not without challenges. Issues such as data sharing constraints, the need for robust regulatory frameworks, and the potential for inequities in access to these advanced technologies must be carefully navigated. Additionally, ensuring the ethical use of AI and addressing concerns around algorithmic bias and transparency are critical to its success. While AI continues to transform the landscape of breast cancer care, a balanced approach that prioritizes inclusivity, fairness, and sustainability in its development and implementation is essential to fully realize its benefits for all populations.

## Figures and Tables

**Figure 1 cancers-17-00197-f001:**
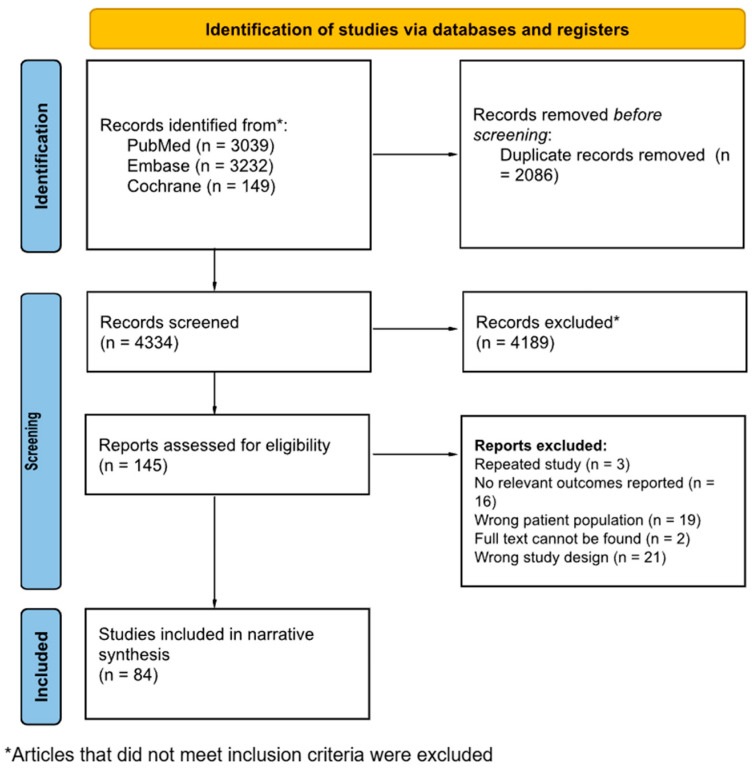
PRISMA flowchart showing the article selection process.

**Figure 2 cancers-17-00197-f002:**
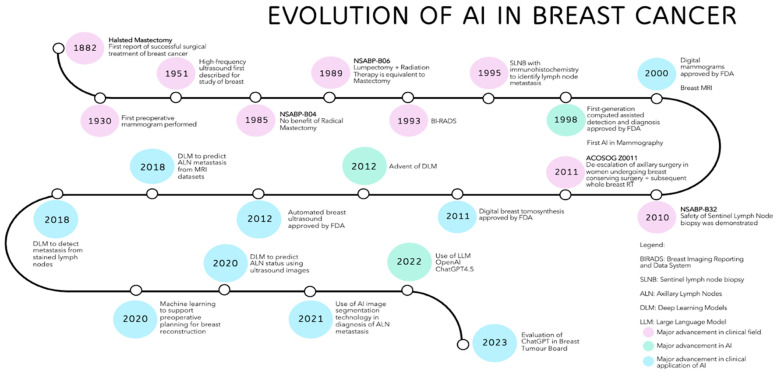
Illustrated timeline summarizing the evolution of AI applications on breast cancer care.

**Table 1 cancers-17-00197-t001:** Key characteristics and findings of the studies reviewed [18,19,20,21,22,23,24,25,26,27,28,29,30,31,32,33,34,35,36,37,38,39,40,41,42,43,44,45,46,47,48,49,50,51,52,53,54,55,56,57,58,59,60,61,62,63,64,65,66,67,68,69,70,71,72,73,74,75,76,77,78,79,80,81,82,83,84,85,86,87,88,89,90,91,92,93,94,95,96,97,98,99,100,101].

Thematic Focus	Author	Study Design	Sample Size	AI Technology	Main Findings
**AI for Breast Cancer Screening**	Aslam et al., 2023 [20]	Prospective	460	AI	AI solution demonstrated high performance in preclassifying cases with a high likelihood for cancer (0.99 AUC, 99% NPV, and 84% PPV) and invasive carcinoma (91% PPV).
**AI for Breast Cancer Screening**	Berg et al., 2023 [23]	Prospective	758	AI-supported US	AI-supported US accurately classified malignancies and benign masses with 0.95 AUC.
**AI for Breast Cancer Screening**	Burger et al., 2023 [24]	Prospective	33	DL	DL-based anomaly detection can help to adjust risk assessment for future breast lesions.
**AI for Breast Cancer Screening**	Byng et al., 2022 [25]	Retrospective	2396	Vara AI system	At 99.0% specificity, AI correctly localized 27.5% of false negatives, 12.2% of missed cases, and 21.1% of advanced/metastatic cancers on prior mammograms, with increased detection for lower-grade carcinomas and those with lymph node involvement. AI identified future malignancies in 2.8% of true interval cancers on screening mammograms.
**AI for Breast Cancer Screening**	Cole et al., 2014 [29]	Retrospective	300	iCAD SecondLook + R2 ImageChecker Cenova	The application of CAD had no statistically significant effect on radiologist AUC, sensitivity, or specificity performance.
**AI for Breast Cancer Screening**	Dembrower et al., 2023 [36]	Prospective	55,581	Insight MMG AI system	AI use in initial mammogram readings increased breast cancer detection rates when implemented in actual screening workflows.
**AI for Breast Cancer Screening**	Dontchos et al., 2022 [37]	Prospective	85,124	DL	The DL model improves mammogram density assessments, optimizing supplemental screening use and enhancing clinical risk models.
**AI for Breast Cancer Screening**	Fan et al., 2020 [41]	Prospective	364	3D-Mask region-based CNN	3D-Mask region-based CNN has advantages over 2D-based mass detection on both the whole data and subgroups with different characteristics.
**AI for Breast Cancer Screening**	Gao et al., 2019 [42]	Review	NR	DL CAD	CAD development for breast imaging is undergoing a paradigm shift based on vast improvement of computing power and rapid emergence of advanced deep learning algorithms, heralding new systems that may hold real potential to improve clinical care.
**AI for Breast Cancer Screening**	He et al., 2023 [48]	Prospective	313	DL CAD	DL-CAD significantly improved radiologists’ diagnostic performance, showing particular potential to reduce the frequency of benign breast biopsies.
**AI for Breast Cancer Screening**	Huang et al., 2023 [49]	Prospective	603	DL	The DLRP can effectively distinguish between luminal and non-luminal breast cancers at early stages before surgery based on pretherapeutic US images and biopsy H&E-stained WSIs, providing a tool to facilitate treatment decision-making in early-stage breast cancers.
**AI for Breast Cancer Screening**	Huo et al., 2021 [51]	Prospective	1345	ML	Compared with clinician diagnosis, the ML model has better diagnostic efficiency, making it potentially useful to assist in screening, particularly in lower-level medical institutions.
**AI for Breast Cancer Screening**	Krajnc et al., 2020 [53]	Prospective	170	ML	ML models based on radiomic tumor features with utilization of data preparation approaches identified aggressive breast cancers with high accuracy.
**AI for Breast Cancer Screening**	Lång et al., 2023 [54]	Prospective	80,033	DL CNN: Transpara	AI-supported mammography screening resulted in a similar cancer detection rate compared with standard double reading, with a substantially lower screen-reading workload, indicating that the use of AI in mammography screening is safe.
**AI for Breast Cancer Screening**	Larsen et al., 2022 [55]	Retrospective	122,969	CNN: Transpara	The proportion of screen-detected cancers not selected by the AI system at the three evaluated thresholds was less than 20%. The overall performance of the AI system was promising according to cancer detection.
**AI for Breast Cancer Screening**	Liu et al., 2023 [60]	Prospective	1760	ML	The proposed ML-based unenhanced radiomics model can effectively identify cases of BC with a diagnostic accuracy similar to that of interpretations provided by the radiologists.
**AI for Breast Cancer Screening**	Mansour et al., 2021 [66]	Prospective	1180	Lunit INSIGHT MMG	AI-mammogram combination could be used as a one setting method to discriminate between cases that require further imaging or biopsy from those that need only time interval follow-up.
**AI for Breast Cancer Screening**	Mckinney et al., 2020 [67]	Retrospective	1511	CAD	The AI system outperformed radiologists on a clinically relevant task of BC identification.
**AI for Breast Cancer Screening**	Mutasa et al., 2020 [68]	Prospective	140	CNN	The CNN algorithm distinguished pure atypical ductal hyperplasia from ductal carcinoma in situ using mammographic images with high specificity.
**AI for Breast Cancer Screening**	Ng et al., 2023 [70]	Prospective	15,953	Mia version 2.0	BC cases detected by AI system and human readers are likely to have similar clinical courses and outcomes, with limited or no downstream effects on screening programs.
**AI for Breast Cancer Screening**	Pöhlmann et al., 2017 [76]	Retrospective	40	CAD	The location weighting scheme for detection of BC masses can be fully automated by using outputs from a CAD algorithm.
**AI for Breast Cancer Screening**	Qian et al., 2021 [77]	Prospective	141	ResNet-18 + SENet	The AI system can be used to assess breast lesions at a level comparable to that of experienced human experts in a prospective setting.
**AI for Breast Cancer Screening**	Rodriguez et al., 2019 [79]	Retrospective	2652	DL CNN: Transpara	The evaluated AI system achieved a cancer detection accuracy comparable to an average breast radiologist in this retrospective setting.
**AI for Breast Cancer Screening**	Sechopoulos et al., 2020 [81]	Review	NR	CAD	CAD systems for detecting BC in screening images are promising due to improved performance and reduced false positives, but their real-world efficacy and optimal use require validation in large-scale trials, alongside addressing medico-legal and ethical issues.
**AI for Breast Cancer Screening**	Shen et al., 2023 [83]	Prospective	360	AI-assisted US	AI-assisted US has good identification ability for breast lesions in Chinese women.
**AI for Breast Cancer Screening**	Sun et al., 2021 [88]	Prospective	5746	Faster R-CNN	The AI model exhibits high accuracy for detecting and diagnosing breast lesions, improves diagnostic accuracy, and saves time.
**AI for Breast Cancer Screening**	Wang et al., 2023 [90]	Prospective	2202	AI-IRT	AI-IRT was highly accurate and less reliant on manual interpretations, with potential for improving pre-clinical screening and increasing screening rates.
**AI for Breast Cancer Screening**	Wei et al., 2022 [91]	Prospective	901	AI-assisted US	CAD software on ultrasound can be used as an effective auxiliary diagnostic tool for differential diagnosis of benign and malignant breast masses and reducing unnecessary biopsy.
**AI for Breast Cancer Screening**	Yang et al., 2024 [93]	Prospective	218	DL	DL model was proven to be more accurate for predicting malignancy in the BI-RADS US 4A breast lesions of patients with dense breasts.
**AI for Breast Cancer Screening**	Ye et al., 2022 [94]	Prospective	66	Dilated deep CCN	The presented dilated deep CNN network has the potential to segment the mass regions in DBT images accurately.
**AI for Breast Cancer Screening**	Yu et al., 2021 [95]	Prospective	3623	CNN	The CNN may have high accuracy for classification of US images of breast masses and perform significantly better than human radiologists.
**AI for Breast Cancer Screening**	Zhao et al., 2022 [97]	Prospective	757	S-Detect	Compared with the conventional US, S-Detect presented higher overall accuracy and specificity.
**AI for Breast Cancer Screening**	Zhou et al., 2023 [99]	Prospective	130	DL	The DL network achieved high and consistent prediction performance and can be implemented without requiring exact tumor segmentation.
**AI for Breast Cancer Screening**	Zhou et al., 2024 [100]	Prospective	486	DNN	The DNN enabled overall b-value DW-MRI to achieve the predictive performance comparable to DCE-MRI.
**AI for Image-detection of Nodal Status**	Sun et al., 2020 [86]	Prospective	479	CNN	CNNs showed numerically better overall performance compared with radiomics models in predicting axillary LN metastasis in BC.
**AI for Image-detection of Nodal Status**	Sun et al., 2022 [87]	Retrospective	169	CNN	DL can be used to predict axillary lymph node metastasis from US images to aid nodal staging in BC.
**AI for Image-detection of Nodal Status**	Zhang et al., 2021 [96]	Retrospective	90	Back-propagation neural network	Back-propagation neural network AI algorithm demonstrated high accuracy, sensitivity, and specificity in US image segmentation, resulting in improved diagnosis of BC axillary lymph node metastasis.
**AI-assisted Histopathology**	Basavanhally et al., 2010 [21]	Retrospective	41	CAD	A CADx system was created to detect and stratify lymphocytic infiltration (LI) in HER2+ breast cancer histopathology images, utilizing architectural features for automatic detection and grading, successfully isolating LI from surrounding tissues and suggesting prognostic implications for disease survival and patient outcomes.
**AI-assisted Histopathology**	Bejnordi et al., 2017 [22]	Retrospective	399	DL models	In a challenge competition, 7 DL algorithms outperformed 11 pathologists in a simulated time-constrained diagnostic setting, with the best algorithm achieving an area under the curve of 0.994, compared to 0.884 for the best pathologist.
**AI-assisted Histopathology**	Chen et al., 2023 [27]	Prospective	795	DL	A DL prognostic model helped further risk stratify HR+ and LN+ BC patients using only H&E slides.
**AI-assisted Histopathology**	Couture et al., 2018 [31]	Retrospective	571	DL	DL accurately predicted tumor grade, estrogen receptor status, Basal-like status, ductal vs. lobular type, and risk of recurrence score from histologic images, suggesting potential for image-based prediction of molecular markers and guiding genomic testing.
**AI-assisted Histopathology**	El et al., 2022 [39]	Prospective	116	CNN: Resnet50 + Xception	CNN models achieved 88% overall accuracy and 95% sensitivity in carcinoma detection, enhancing precision in diagnostic pathology workflows for breast cancer.
**AI-assisted Histopathology**	Elsharawy et al., 2021 [40]	Retrospective	612	CNN	An AI grade, based on distinctive nucleolar prominence features, effectively communicates patient outcome information across three independent cohorts, offering potential for gene discovery and providing valuable second opinions.
**AI-assisted Histopathology**	Han et al., 2017 [46]	Retrospective	82	Class structure-based deep CNN	The novel class structure-based deep CNN model demonstrated reliable and accurate multi-class BC classification in histopathological images, outperforming traditional models and proving stable across different magnifications, thus holding significant potential for clinical diagnosis and addressing the shortage of professional pathologists.
**AI-assisted Histopathology**	Li et al., 2022 [58]	Retrospective	300	AI	AI shows satisfactory inter-observer repeatability and accuracy for Ki-67 label index reproducibility, suggesting it as the preferred method, while standard reference card remains a standard candidate until AI is widely adopted.
**AI-assisted Histopathology**	Liu et al., 2018 [60]	Retrospective	399	DL model: LYmph Node Assistant	AI algorithms can exhaustively evaluate every tissue patch on a slide, achieving higher tumor-level sensitivity than, and comparable slide-level performance to, pathologists.
**AI-assisted Histopathology**	Nateghi et al., 2021 [69]	Retrospective	500	CNN	A fully automated cancer grading system is introduced utilizing mitosis counting from whole slide images, comprising three automated subsystems for mitosis activity region detection, mitosis detection, and tumor proliferation grading.
**AI-assisted Histopathology**	Pesapane et al., 2021 [73]	Retrospective	100	ITK-SNAP + PyRadiomics	The integration of MRI-based radiomic features with clinical and biological data slightly enhances pre-treatment prediction of pCR to neoadjuvant chemotherapy in BC.
**AI-assisted Histopathology**	Pfob et al., 2020 [74]	Prospective	570	CNN	Safely selected patients without residual disease as assessed by the CNN algorithm may be spared from breast surgery in future trials.
**AI-assisted Histopathology**	Pfob et al., 2021 [75]	Prospective	478	ML: TensorFlow + Keras	A multivariate algorithm can accurately select breast cancer patients without residual cancer after neoadjuvant treatment.
**AI-assisted Histopathology**	Saltz et al., 2018 [80]	Retrospective	4612	DL CNN	A DL-derived “computational stain” was developed to identify tumor-infiltrating lymphocytes (TILs) in standard pathology cancer images, processing 5202 digital images from 13 cancer types.
**AI-assisted Histopathology**	Steiner et al., 2018 [85]	Retrospective	70	DL model: LYmph Node Assistant	Combining DL algorithms with the expertise of pathologists may offer enhanced diagnostic performance, as pathologists can contextualize clinical implications and optimize diagnosis while algorithms can highlight critical information.
**AI-assisted Histopathology**	Whitney et al., 2018 [92]	Retrospective	178	CNN	Computerized image analysis of digitized H&E pathology images of early-stage ER+ BC could potentially predict the corresponding Oncotype DX risk categories.
**AI-assisted Histopathology**	Zhang et al., 2024 [101]	Prospective	129	DL	The combination of D-FFOCT imaging with a deep learning algorithm has the potential to simplify intraoperative cancer diagnosis.
**AI-assisted Histopathology**	Zheng et al., 2023 [98]	Prospective	1912	RefineNet + Xception	The AI model demonstrated the potential for the segmentation and classification of breast lesions and had good generalization ability and clinical applicability.
**AI in assessing post-neoadjuvant chemotherapy (NACT) response**	Dammu et al., 2023 [32]	Prospective	155	DL-CNN	DL-CNN model can be used to identify patients who are likely to respond to neoadjuvant chemotherapy at diagnosis or early treatment for future clinical decision-making using MRI.
**AI in assessing post-neoadjuvant chemotherapy (NACT) response**	Dasgupta et al., 2024 [34]	Prospective	60	ML	ML-assisted US-radiomics accurately predicted final neoadjuvant chemotherapy treatment response with 97% accuracy.
**AI in assessing post-neoadjuvant chemotherapy (NACT) response**	De Sanctis et al., 2024 [35]	Prospective	105	ML	The integrated ML model predicted pCR with an AUC of 0.79 ± 0.09 using pre-treatment microbiota and [18] F-FDG PET/CT radiomic features.
**AI in assessing post-neoadjuvant chemotherapy (NACT) response**	El et al., 2019 [38]	Prospective	42	DL-CNN	The CNN model achieved 92.72% accuracy and a 96% AUC using 20% of 3D validation data in predicting tumor response to chemotherapy
**AI in assessing post-neoadjuvant chemotherapy (NACT) response**	Gu et al., 2022 [45]	Prospective	168	DL	DL radiomics holds promise for effectively predicting NAC response at its early stage for BC patients.
**AI in assessing post-neoadjuvant chemotherapy (NACT) response**	Huang et al., 2023 [49]	Prospective	255	DL CNN	The pretreatment CNN model based on the dual-modal US and molecular data achieved excellent performance for predicting the response to chemotherapy in breast cancer.
**AI in assessing post-neoadjuvant chemotherapy (NACT) response**	Khan et al., 2022 [52]	Review	NR	CNN	While DL shows promise in predicting treatment response, challenges include data size, model comparison, and integration of diverse MRI data, requiring further validation and refinement for clinical use.
**AI in assessing post-neoadjuvant chemotherapy (NACT) response**	Li et al., 2023 [59]	Prospective	138	Sklearn + FeAture Explorer	The pCR of axillary LNs in breast cancer following neoadjuvant chemotherapy can be precisely predicted using CT-based radiomics.
**AI in assessing post-neoadjuvant chemotherapy (NACT) response**	Liu et al., 2023 [60]	Prospective	189	DL	An independent predictive model built on combining deep learning technology, CT images and PD-L1 expression could identify advanced breast cancer patients most likely to benefit from immunotherapy.
**AI in assessing post-neoadjuvant chemotherapy (NACT) respons**	Tahmassebi et al., 2019 [89]	Prospective	38	XGBoost	ML enables early prediction of pCR to NAC as well as survival outcomes in breast cancer patients with high accuracy.
**AI in breast cancer margin assessment**	Levy et al., 2023 [56]	Retrospective	151	DL CNN	The DL model’s accurate identification of 96.8% pathology-positive margins demonstrates the clinical potential of AI-enhanced margin visualization in breast cancer surgery, potentially lowering reoperation rates due to residual tumors.
**AI as a Clinical Decision Support Tool**	Al-Hilli et al., 2023 [19]	Prospective	39	Automated Chatbot	Patients undergoing pre-test genetic counseling using an automated chatbot had similar satisfaction and comprehension to in-person genetic testing.
**AI as a Clinical Decision Support Tool**	Chaix et al., 2019 [26]	Prospective	4737	Automated Chatbot	The chatbot (Vik) enhanced support for breast cancer patients and increased medication adherence rates.
**AI as a Clinical Decision Support Tool**	Choi et al., 2023 [28]	Retrospective	2931	ChatGPT	Efficient prompts for LLMs facilitated the extraction of crucial information from extensive medical records, demonstrating the potential of natural language processing with LLMs in BC patient care.
**AI as a Clinical Decision Support Tool**	Comes et al., 2021 [30]	Prospective	158	CNN	CNN features extracted from pre-treatment and early-treatment exams were revealed to be strong predictors of BC recurrence.
**AI as a Clinical Decision Support Tool**	Dasgupta et al., 2021 [33]	Prospective	83	ML	ML-assisted US-radiomic features obtained before start of treatment can predict risk of disease recurrence with reasonable accuracy.
**AI as a Clinical Decision Support Tool**	Garberis et al., 2022 [43]	Prospective	1429	DL	DL model provided additional prognostic information as compared to current clinico-pathological factors.
**AI as a Clinical Decision Support Tool**	Griewing et al., 2023 [44]	Retrospective	20	ChatGPT 3.5	There was a 50% concordance between the LLM and multidisciplinary tumor board for patient profiles, including precancerous lesions, and a 58.8% concordance for invasive breast cancer profiles. However, due to occasional fraudulent decisions by the LLM, the current development status of publicly available LLMs is deemed insufficient as a support tool for tumor boards.
**AI as a Clinical Decision Support Tool**	Haver et al., 2023 [47]	Retrospective	NR	ChatGPT	ChatGPT shows promise in automating the dissemination of accurate healthcare information regarding breast cancer prevention and screening, yet physician oversight is essential due to potential for inappropriate and unreliable responses.
**AI as a Clinical Decision Support Tool**	Li et al., 2020 [57]	Prospective	121	DL generative adversarial network model	Computer-extracted nuclear histomorphometric features were associated with disease-free survival in ductal carcinoma in-situ patients.
**AI as a Clinical Decision Support Tool**	Liu et al., 2024 [61]	Prospective	122,508	ML risk prediction model	BC risk prediction models with greater discrimination accuracy than existing state-of-the-art methods were constructed for the precise screening of groups with a high risk of developing BC in China.
**AI as a Clinical Decision Support Tool**	Lukac et al., 2023 [102]	Prospective	10	ChatGPT	ChatGPT has the potential to find its spot in clinical medicine, but the current version is not able to provide specific recommendations for the therapy of patients with primary BC.
**AI as a Clinical Decision Support Tool**	Ma et al., 2023 [64]	Prospective	230	High-Resolution Net	AI can be used to predict normal lung dose in adjuvant radiotherapy following breast-conserving surgery for invasive BC.
**AI as a Clinical Decision Support Tool**	Manikis et al., 2023 [65]	Prospective	706	BOUNCE-CDS	The BOUNCE CDS tool paves the way for personalized risk assessment methods to identify patients at high risk of adverse well-being outcomes and direct valuable resources toward those most in need of specialized psychological interventions.
**AI as a Clinical Decision Support Tool**	Park et al., 2019 [71]	Prospective	246	ML	ML using low-dose perfusion CT is a useful noninvasive tool for predicting prognostic biomarkers and molecular subtypes of invasive BC.
**AI as a Clinical Decision Support Tool**	Park et al., 2021 [72]	Prospective	147	ML	ML using texture and perfusion characteristics of breast cancer with low-dose CT has potential value for predicting prognostic factors and risk stratification in BC.
**AI as a Clinical Decision Support Tool**	Rao et al., 2023 [78]	Retrospective	NR	ChatGPT 4.0, ChatGPT 3.5	There is potential feasibility of employing LLMs like ChatGPT for radiologic decision-making, which could enhance clinical workflow and optimize the utilization of radiology services. Further exploration of use cases and improved accuracy are essential for evaluating and effectively integrating such tools.
**AI as a Clinical Decision Support Tool**	Shamai et al., 2022 [82]	Prospective	3376	CNN	The system was able to predict the expression of PD-L1 and PD-1 in all experiments and could serve as a decision support and quality assurance in clinical practice.
**AI as a Clinical Decision Support Tool**	Sorin et al., 2023 [84]	Retrospective	10	ChatGPT	ChatGPT, as a clinical decision support tool in breast tumor board decisions, achieved 70% agreement with board recommendations, with strengths including concise case summaries and explanations, but had the lowest scores on clinical recommendations. Limitations include small sample size, biased training data, legal and ethical concerns, and cybersecurity risks

Legend: AI = Artificial Intelligence; DL = deep learning; ML = machine learning; CNN = convolutional neural network; DNN = Deep Neural Network; CAD = computer-aided diagnosis; US = ultrasound; MRI = Magnetic Resonance Imaging; DCE = dynamic contrast-enhanced; DW = diffuse-weighted; AUC = area under receiver operating characteristic curve; NPV = negative predictive value; PPV = positive predictive value; H&E = Hematoxylin and Eosin; HR = hormone receptor; ER = estrogen receptor; LN = lymph node; BC = breast cancer; pCR = pathological complete response; OCT = optical coherence tomography; LLM = Large Language Model; NR = Not Reported.

## Data Availability

Data sharing is not applicable to this article as no new data were created or analyzed in this study. The data that support the findings of review are derived from publicly available sources, including databases such as PubMed, Embase, and the Cochrane Library.

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
