# Peer review of "Harnessing Artificial Intelligence to Enhance Global Breast Cancer Care: A Scoping Review of Applications, Outcomes, and Challenges"

_cancers, 2025, doi:10.3390/cancers17020197_

Round 1

Reviewer 1 Report

Comments and Suggestions for Authors

The works concerning the role and place of artificial intelligence methods and systems in breast cancer care are reviewed. Clear and vivid picture of current state and possible ways of development is presented. The work has some minor flaws, that are:

1. Lines 156-160 describe the division of the Table 1 into themes. It is better to place this description before the table itself and additionally add sub-titles into the table. Now reading the table first produces a confusion: why the sort order of the table by first author is violated several times?

2. Line 156 says that six themes were "generated". Clustering knowledge base into topics is always disputable matter. This is the clue issue in understanding and describing the field. So, it is necessary to explain how the themes were "generated". By human expert? Automatically? Using some formal rule?

3. Figure 3 depicts the stages of patients journey in a healthcare system, which are supported by AI implementations. It is advisable to provide references to the appropriate literature for each of the stages. Maybe in the form of the table.

After correcting this issues the paper can be published.

Author Response

Comment 1: Lines 156-160 describe the division of Table 1 into themes. It is better to place this description before the table itself and additionally add sub-titles into the table. Now reading the table first produces confusion: why the sort order of the table by first author is violated several times?

Response 1: Thank you for the comment. We agree with the reviewer that placing the description of the themes before the table will enhance clarity. We have moved the text in Lines 156–160 to precede Table 1 and have now reorganized Table 1 according to the themes.

Comment 2: Line 156 says that six themes were "generated". Clustering knowledge base into topics is always a disputable matter. This is the key issue in understanding and describing the field. It is necessary to explain how the themes were "generated". By human expert? Automatically? Using some formal rule?

Response 2: Thank you for the comment. In our methods section, we have now clearly specified that the themes were generated through a manual narrative synthesis by two researchers, supported by consensus discussions with senior authors, “Two independent researchers (J.L.L.C. and G.S.H.) manually reviewed the extracted data to identify recurring patterns and topics, and the process involved iterative discussion and consensus building with the senior authors (Q.X.N. and S.S.N.G.) to name and label the themes. The charting table facilitated a thematic synthesis of the results, allowing us to identify key trends, gaps, and variations in AI applications for breast cancer care.”

Comment 3: Figure 3 depicts the stages of patients' journey in a healthcare system, which are supported by AI implementations. It is advisable to provide references to the appropriate literature for each of the stages. Maybe in the form of a table.

Response 3: Thank you for the suggestion. We have now provided references directly within Figure 3.

We are immensely grateful to Reviewer 1 for all the valuable suggestions and comments.

Reviewer 2 Report

Comments and Suggestions for Authors

This manuscript explore the transformative potential of AI in breast cancer care. While the breadth of topics covered is commendable, the synthesis of findings lacks critical depth in some sections, such as the application of AI in low-resource settings, which could benefit from greater contextualization. Moreover, the methodological rigor in data extraction and thematic categorization is not thoroughly justified, leaving ambiguities in reproducibility. Expanding on implementation barriers and patient-centric outcomes would enhance the manuscript’s relevance. Addressing these areas through more nuanced discussion and integrating broader global perspectives would significantly elevate the manuscript's quality.

Comments:

1. The methodological framework used for thematic synthesis (Popay et al.) is robust but inadequately contextualized for AI applications in healthcare. For example, the rationale for including grey literature lack explanation, reducing methodological transparency. The authors state, “A grey literature search was conducted using Google…” but omit details on how this ensured the inclusion of relevant high-quality studies. Providing a stronger justification for this choice and specifying inclusion criteria would improve the methodological clarity and the validity of the conclusions drawn.

2. The discussion of challenges related to data quality and algorithm transparency is insufficiently detailed. For instance, while noting "persistent challenges in AI adoption," the manuscript fails to analyze ethical considerations like data bias or disparities in algorithmic performance across populations. A deeper exploration of such issues, supported by relevant case studies or comparative frameworks, would enhance the critical examination. Additionally, key references are overly generalized, such as [123], which does not address AI ethics explicitly, undermining the depth of discussion.

3. The claim that AI reduce global disparities is overgeneralized. For example, the statement, “AI has the potential to bridge gaps,” lacks empirical evidence or citation of region-specific studies. Integrating quantitative data from underserved regions is essential for substantiation.

4. The manuscript repeatedly mentions “streamlining workflows” without clarifying how this benefit low-resource settings. Specific examples of reduced clinician workload or improved outcomes in underserved areas would strengthen the argument’s practical applicability and generalizability.

5. The authors’ focus on imaging technologies ignores emerging AI applications in patient education or mental health support, e.g., chatbots. Addressing such uses would diversify the scope and align with broader healthcare goals beyond diagnostic accuracy.

6. Reference [20] discusses diagnostic competition but is cited ambiguously in the context of AI validation. Clarifying its relevance to algorithmic sensitivity and specificity in detecting micrometastasis would bolster the manuscript’s methodological coherence.

Author Response

Comment 1: The methodological framework used for thematic synthesis (Popay et al.) is robust but inadequately contextualized for AI applications in healthcare. For example, the rationale for including grey literature lack explanation, reducing methodological transparency. The authors state, “A grey literature search was conducted using Google…” but omit details on how this ensured the inclusion of relevant high-quality studies. Providing a stronger justification for this choice and specifying inclusion criteria would improve the methodological clarity and the validity of the conclusions drawn.

Response 1: Thank you for the comments. We have now further elaborated on the choice of the framework for narrative synthesis by Popay et al. (2006). Popay et al.’s framework provides important guidance on the conduct of narrative synthesis and it has been used elsewhere to analyze AI applications in healthcare (reference: Abbasgholizadeh Rahimi S, Cwintal M, Huang Y, Ghadiri P, Grad R, Poenaru D, Gore G, Zomahoun HTV, Légaré F, Pluye P. Application of Artificial Intelligence in Shared Decision Making: Scoping Review. JMIR Med Inform. 2022;10(8):e36199. doi: 10.2196/36199).

Regarding the inclusion of a grey literature search, we have now further explained that a grey literature search was conducted using Google to identify preprints and unpublished studies with substantial data, specifically non-peer-reviewed evidence, such as government reports, organizational white papers, or preprint studies, which may be significant in the rapidly evolving field of AI in healthcare. This approach aimed to capture emerging insights and ongoing research not yet published in traditional journals. Preprints with robust data and relevance to AI in breast cancer care were included in the final review to ensure a comprehensive synthesis of the field.

Comment 2: The discussion of challenges related to data quality and algorithm transparency is insufficiently detailed. For instance, while noting "persistent challenges in AI adoption," the manuscript fails to analyze ethical considerations like data bias or disparities in algorithmic performance across populations. A deeper exploration of such issues, supported by relevant case studies or comparative frameworks, would enhance the critical examination. Additionally, key references are overly generalized, such as [123], which does not address AI ethics explicitly, undermining the depth of discussion.

Response 2: Thank you for the comment. We have now added further discussion (with appropriate supporting references) detailing specific examples of AI ethical challenges, such as racial bias in facial recognition algorithms or disparities in healthcare outcomes due to underrepresentation of certain populations in training data.

Comment 3: The claim that AI reduce global disparities is overgeneralized. For example, the statement, “AI has the potential to bridge gaps,” lacks empirical evidence or citation of region-specific studies. Integrating quantitative data from underserved regions is essential for substantiation.

Response 3: Thank you for the comment. We agree that the claim requires additional evidence to avoid overgeneralization. As such, we have now added further region-specific studies and quantitative data illustrating AI’s role in addressing healthcare disparities, particularly in low-resource settings.

Comment 4: The manuscript repeatedly mentions “streamlining workflows” without clarifying how this benefit low-resource settings. Specific examples of reduced clinician workload or improved outcomes in underserved areas would strengthen the argument’s practical applicability and generalizability.

Response 4: Thank you for the comment. In our discussion section, we have now provided further description of specific examples of AI implementations that would reduce clinician workload, such as automated triage systems and AI-assisted imaging in primary care settings.

Comment 5: The authors’ focus on imaging technologies ignores emerging AI applications in patient education or mental health support, e.g., chatbots. Addressing such uses would diversify the scope and align with broader healthcare goals beyond diagnostic accuracy.

Response 5: Thank you for the comment. We have now further addressed and added discussion around the use of patient education chatbots as suggested.

Comment 6: Reference [20] discusses diagnostic competition but is cited ambiguously in the context of AI validation. Clarifying its relevance to algorithmic sensitivity and specificity in detecting micrometastasis would bolster the manuscript’s methodological coherence.

Response 6: Thank you for the comment. We have further elaborated upon the findings of the study by Ehteshami et al. (2017), “The study by Ehteshami et al. [20] provides a comprehensive evaluation of deep learning algorithms for the detection of lymph node metastases in women with breast cancer. This study illustrates the importance of sensitivity and specificity metrics in validating AI models, demonstrating that several deep learning algorithms outperformed human pathologists in detecting micrometastases under time-constrained conditions (sensitivity up to 92% and AUC of 0.994). These findings underscore the potential of AI to complement clinical workflows while also highlighting the rigorous validation needed to ensure their reliability in real-world applications.”

We are immensely grateful to Reviewer 2 for all the valuable suggestions and comments.

Round 2

Reviewer 1 Report

Comments and Suggestions for Authors

The authors have answered all of the issues of the previous comments and revised the paper accordingly. It can be published now.

Author Response

Thank you for the kind words and encouragement!

Reviewer 2 Report

Comments and Suggestions for Authors

The latest version of the manuscript indicates that some issues have been partially addressed, while others remain unresolved or require further development.

Below is an analysis of the specific comments:

1. The addition of studies on regional AI adoption disparities (e.g., by citing region-specific success stories) would add critical depth and address global inequity concerns more robustly.  

2. The conclusion section should incorporate more balanced discussion on AI’s limitations, such as data sharing and regulatory challenges, to provide a comprehensive assessment.

Author Response

Comment 1: The addition of studies on regional AI adoption disparities (e.g., by citing region-specific success stories) would add critical depth and address global inequity concerns more robustly.  

Reply 1: Thank you for the suggestion. We have now added in the discussion section that, "As late-stage breast cancer typically has a poorer prognosis compared to early-stage cases [126], the iBreastExam (iBE) is an example of an innovative device leveraging emerging AI technologies, which offers a cost-effective alternative to conventional mammography for screening [127]. It has been positioned as a valuable adjunct for breast cancer detection in LMICs with constrained healthcare budgets; in a study conducted in India involving 989 healthy women undergoing annual health checks, iBE demonstrated a 19% higher sensitivity than traditional clinical breast examination for detecting breast lesions, along with impressive specificity (94%) and negative predictive value (98%) [128]. However, despite such success stories of AI in breast cancer screening, realizing their full potential and seamlessly integrating into LMIC settings demands careful consideration of multiple factors."

Comment 2: The conclusion section should incorporate more balanced discussion on AI’s limitations, such as data sharing and regulatory challenges, to provide a comprehensive assessment.

Reply 2: Thank you for the comment. We have noted your comment and revised the conclusion accordingly, "In summary, AI has demonstrated a wide range of clinical applications in breast cancer care, including diagnostic imaging, histopathology, treatment planning, and clinical decision support. These advancements offer promising opportunities to address global disparities in healthcare access and improve patient outcomes. However, the widespread adoption of AI is not without challenges. Issues such as data sharing constraints, the need for robust regulatory frameworks, and the potential for inequities in access to these advanced technologies must be carefully navigated. Additionally, ensuring the ethical use of AI and addressing concerns around algorithmic bias and transparency are critical to its success. While AI continues to transform the landscape of breast cancer care, a balanced approach that prioritizes inclusivity, fairness, and sustainability in its development and implementation is essential to fully realize its benefits for all populations."